# OpenReview forum: "Label Transfer Hypothesis: A Clinical Prior Knowledge-Guided Approach for Disease Diagnosis"
_ICLR.cc/2026/Conference — ICLR 2026 Conference Withdrawn Submission_

### Official Review · Reviewer_G7AA · 2025-10-22

**Soundness:** 1
**Presentation:** 1
**Contribution:** 2
**Rating:** 2
**Confidence:** 4

**Summary:**

This paper proposes label transfer hypothesis (LTH), a theoretical framework designed for addressing limited labeled data in medical image classification. This model transfers knowledge from segmentation tasks to classification tasks, integrating clinical diagnosis and treatment logic to improve diagnosis performance.

The proposed LTH is evaluated on diabetic macular edema and retinal vein occlusion classificatin tasks, and compared against a few supervised baselines.

**Strengths:**

* S1: This paper explores weakly/self-supervised learning for limited labeled data, which is of interest to the community.

**Weaknesses:**

**Major Weakness**

* W1: The motivation for using segmentation datasets to address limited classification labels is questionable, since medical image segmentation annotations are usually require more effort and segmentatioin datasets are often smaller than classification datasets (e.g., the IDRiD dataset vs. Retinal Vessel segmentation combined dataset in Tab. 1).

* W2: The “theoretical framework” lacks any formal theory, equations, or analysis to justify the claim.

* W3: The related works section is limited, lacking discussion of existing work on weakly/unsupervised medical image analysis [R1, R2].

* W4: The experimental comparisons are limited. This paper does not include comparisons with SOTA models designed for learning with limited labeles.

* W5: Writing and organization need major improvement. Current methodology section is hard to follow and cannot distinguish the methodological novelty. The authors could consider provide some equations to help with.

**Minor Weakness**

* MW1: Some references are not cited correctly, e.g., Line 125

* MW2: What are the experimental setups of compared baselines?

[R1] Atwany, Mohammad Z., Abdulwahab H. Sahyoun, and Mohammad Yaqub. "Deep learning techniques for diabetic retinopathy classification: A survey." IEEE Access 10 (2022).

[R2] Raza, Khalid, and Nripendra K. Singh. "A tour of unsupervised deep learning for medical image analysis." Current Medical Imaging Reviews (2021).

**Questions:**

**Primary  Questions/Suggestions**

* QS1: What is the motivation for using segmentation datasets to handle limited labeled data in classification tasks?

* QS2: Can the authors formalize the “theoretical framework” and provide equations or analysis?

* QS3: Please include comparisons with SOTA methods for limited labeled data.

* QS4: The authors could consider refining the writing and improving the paper’s structure for clarity.

**Minor Questions/Suggestions**

* MQS1: Please correct the citation issues.

---

### Official Review · Reviewer_8MEX · 2025-10-29

**Soundness:** 2
**Presentation:** 2
**Contribution:** 2
**Rating:** 2
**Confidence:** 4

**Summary:**

The manuscript proposes the LTH as a theoretical framework to address the bottleneck of large-scale, expensive, and time-consuming disease-level annotation in medical image analysis, particularly in ophthalmology. The core idea is to leverage precise lesion segmentation (a pixel-level, clinically-guided task) as "implicit diagnostic labels" to enable knowledge transfer to the image-level disease classification task, thereby eliminating the need for explicit, supervised disease labels. The proposed method is validated on DME and RVO classification tasks and achieves high accuracy (86.00% for DME, 86.21% for RVO) without requiring traditional disease-level annotations, which the authors claim is comparable or superior to conventional supervised methods.

**Strengths:**

1. The LTH proposal is an interesting idea that tries to bridge the gap between pixel-level segmentation and image-level classification. This moves beyond treating segmentation merely as an auxiliary task to establishing it as the fundamental diagnostic evidence.
2. The entire methodology is consistent with clinical diagnostic logic—identifying specific pathological features before synthesizing them into a diagnosis.
3. Achieving compelling DME and RVO classification results even without any disease-level image labels.

**Weaknesses:**

1. The experimental validation in the paper was conducted on two very small and highly balanced classification tasks, where the methods presented in Table 1 merely employed basic supervised approaches—specifically, supervised three-grading for RVO and MobileNet/ResNet18 for DME. However, it is essential to benchmark the proposed approach against state-of-the-art supervised methods on large-scale datasets to convincingly demonstrate its performance parity or superiority, rather than relying on comparisons with outdated or suboptimal models.
2. The manuscript frequently asserts a "Theoretical foundation for clinical interpretability," yet this pivotal claim remains largely unsubstantiated within the text. Furthermore, the core approach of leveraging lesion information to inform image-level classification is a conventional methodology, routinely employed in fields such as Whole Slide Imaging (WSI) for cellular pathology, where lesional cells are first detected before aggregating this evidence for a final diagnosis. Consequently, the methodological contribution appears limited.
3. Some implementation details are insufficiently clarified, specifically regarding the selection process of the segmented regions during clinical feature extraction. Moreover, the paper also contains a few typographical errors, such as incorrect citations.
4. A critical evaluation is necessary regarding whether the presented experimental results originate solely from a meticulously curated, small-scale dataset. The authors must disclose the performance metrics achieved when the methodology is benchmarked on the original, established public datasets.

**Questions:**

1. The training regimen of this paper centers predominantly on the segmentation phase; therefore, the quantitative performance metrics for the segmentation task are essential and should be explicitly reported.
2. How about the performance metrics achieved on the original, established public datasets?

---

### Official Review · Reviewer_x3fx · 2025-10-29

**Soundness:** 3
**Presentation:** 3
**Contribution:** 2
**Rating:** 2
**Confidence:** 5

**Summary:**

This paper proposes the Label Transfer Hypothesis (LTH) to address annotation scarcity. The key idea is that precise lesion segmentation, guided by clinical logic, can serve as implicit diagnostic labels for disease classification, enabling knowledge transfer from segmentation to classification tasks.

**Strengths:**

1. The paper is clearly written, with a well-formulated problem statement.
2. The paper is well-organised.

**Weaknesses:**

1. The lesion-specific encoder selection strategy may be unnecessary and could introduce excessive computational cost and instability during training. A unified deep encoder might already capture sufficient lesion-specific features.
2. The technical novelty is limited; the employed backbone (e.g., VGG encoder) and methodology are somewhat outdated compared to modern architectures.
3. Although the work aims to reduce annotation costs for disease classification, it still relies on pixel-wise segmentation annotations, which are more expensive and time-consuming to obtain than image-level labels.
4. The experimental validation is insufficient, as the paper only compares with simple baselines and their variants, without benchmarking against more established or state-of-the-art methods.

**Questions:**

N/A

---

### Official Review · Reviewer_n8nE · 2025-10-31

**Soundness:** 2
**Presentation:** 2
**Contribution:** 1
**Rating:** 0
**Confidence:** 4

**Summary:**

The paper presents the Label Transfer Hypothesis (LTH), a framework aimed at reducing dependence on large, expert-labeled datasets in medical image diagnosis. It builds on the observation that characteristic lesions in diseases like diabetic macular edema and retinal vein occlusion already encode diagnostic information. By treating these lesion annotations as implicit disease labels, LTH transfers knowledge from segmentation to classification tasks. This approach aligns with how clinicians reason through diagnosis, first identifying pathological features and then synthesizing them into conclusions.

The proposed pipeline involves three steps: lesion segmentation using a modified U-Net with attention mechanisms, feature extraction with a pre-trained VGG19 model and dimensionality reduction, and unsupervised disease classification through clustering of lesion patterns. Experiments show that the method achieves around 86 percent accuracy on both diseases, comparable to supervised models but with far fewer labeled samples. The framework offers a clinically interpretable and resource-efficient alternative for disease classification grounded in medical reasoning.

**Strengths:**

1. The paper addresses a clear and timely problem in medical image analysis: the lack of a unified framework that connects lesion-level segmentation with disease-level diagnosis. Most existing models either depend heavily on large labelled datasets or treat lesions as auxiliary cues rather than the foundation of diagnosis. The authors argue that this gap limits both scalability and clinical interpretability. Their problem statement reframes the task by proposing that lesion annotations, when guided by clinical reasoning, can act as implicit diagnostic labels, offering a way to reduce annotation costs while preserving diagnostic rigour.

2. The system’s design effectively operationalizes this idea through a modular and clinically aligned workflow. It separates the process into three stages: segmentation, feature extraction, and clustering, mirroring how clinicians move from observing individual lesions to forming diagnostic conclusions. This structure allows each module to be trained or refined independently, supports transparent reasoning linked to identifiable lesion patterns, and makes the framework adaptable to other diseases with localized pathological markers. The result is an interpretable and resource-efficient design that bridges computational modelling with clinical practice.

**Weaknesses:**

Major weakness
1. The main weakness is in the lack of true end-to-end learning and the limited validation scope. While the segmentation model is trained, the subsequent feature extraction and clustering steps are unsupervised and rely heavily on hand-crafted choices (e.g., lesion weighting, PCA reduction, clustering algorithm). This breaks the learning pipeline and limits adaptability to more complex or overlapping disease patterns. Additionally, the experiments are restricted to two retinal diseases (DME and RVO) using moderate-sized public datasets, without testing cross-dataset generalization, robustness to noisy annotations, or clinical deployment settings. As a result, it remains uncertain how well the Label Transfer Hypothesis scales to broader modalities or real-world clinical variability.

2. The framework lacks theoretical or quantitative justification for the label transfer assumption itself. While the paper proposes that lesion patterns can serve as implicit diagnostic labels, it does not formally define or test the boundaries of this assumption, such as when lesions are non-specific, overlapping across diseases, or influenced by imaging artifacts. There is no mathematical proof or empirical analysis showing under what conditions the transfer remains valid. This leaves the “Label Transfer Hypothesis” more as an intuitive idea than a rigorously established principle, which may limit its general acceptance or theoretical impact.

Minor - the paper includes various typos and missing references

**Questions:**

1. How sensitive are the results to the specific hand-crafted components, such as lesion importance weights, PCA dimensions, or clustering algorithms?

2. Do you have any evidence that the approach would scale beyond the two retinal diseases studied or to other imaging modalities?

3. Have you conducted any quantitative analysis to test the validity of the label transfer assumption, such as measuring information overlap between lesion maps and disease labels?

4. How does the framework handle cases where lesion patterns are shared across multiple diseases or are non-specific?

---

### Note · Authors · 2026-01-21

I have read and agree with the venue's withdrawal policy on behalf of myself and my co-authors.